# I Think I Can, I Think I Can: Effects of Entrepreneurship Orientation on Entrepreneurship Intention of Saudi Agriculture and Food Sciences Graduates

Ibrahim A. Elshaer [1,2,3,*] and Abu Elnasr E. Sobaih [1,2,4]

1   The Saudi Investment Bank Scholarly Chair for Investment Awareness Studies, The Deanship of Scientific Research, The Vice Presidency for Graduate Studies and Scientific Research, King Faisal University, Al-Ahsa 31982, Saudi Arabia
2   Management Department, College of Business Administration, King Faisal University, Al-Hassa 31982, Saudi Arabia
3   Faculty of Tourism and Hotel Management, Suez Canal University, Ismailia 41522, Egypt
4   Faculty of Tourism and Hotel Management, Helwan University, Cairo 12612, Egypt
*   Correspondence: ielshaer@kfu.edu.sa

**Abstract:** Entrepreneurship plays a significant role in achieving the national agenda and the development of nations. The leadership of the Kingdom of Saudi Arabia (KSA) pays significant attention to the role of entrepreneurship by supporting the Saudi Vision 2030 and its programs, e.g., the National Transformation Program and the Human Capability Development Program. The current research examines the effects of entrepreneurship orientation on entrepreneurship intentions among agriculture and food sciences graduates in KSA. More exactly, the research examines the interrelationship and effects of risk-taking, innovativeness, and pro-activeness on the entrepreneurship intentions of higher education graduates, especially among agriculture and food sciences graduates. It also examines the mediating effect of innovativeness and pro-activeness between entrepreneurship orientation and entrepreneurship intention. For this purpose, a pre-examined questionnaire was collected randomly from 440 graduates at several KSA universities. The results of structural equation modeling (SEM) supported all research hypotheses and confirmed a direct significant effect of risk-taking on the entrepreneurship intention of students and an indirect effect through innovativeness and pro-activeness. The results confirmed complimentary mediation of innovativeness and pro-activeness in the link between entrepreneurship orientation and entrepreneurship intention. The findings of the research offer some implications for scholars and policymakers in the Saudi context, which are discussed.

**Keywords:** entrepreneurship orientation; risk-taking; innovativeness; pro-activeness; entrepreneurship intentions; higher education; Saudi Arabia

## 1. Introduction

Entrepreneurship has gained much attention from decision-makers, researchers, and practitioners due to its contribution to the economy and sustainable development [1]. It also has several positive impacts, e.g., social development [2], technological development [3], enhancing the national economy [4], and competitiveness in national and international markets [5,6]. Earlier studies (e.g., [1,7–10]) showed that entrepreneurship is a driver for economic growth, promoting innovation and business opportunities, which are essential for socio-economic development worldwide. Furthermore, entrepreneurship plays a vital role in the development of youth towards becoming self-reliant [11]. Hence, entrepreneurship contributes to social development through new ventures and job creation not only for entrepreneurs but also for others [12–14]. In this context, a recent study by Tripathi [15] confirmed that small and medium-sized businesses account for approximately 80% of the global GDP and provide employment opportunities globally.

Policymakers all over the world have recognized the importance of entrepreneurship. Hence, they are increasingly investing in the small and medium-sized enterprises (SME) sector to create jobs and promote socio-economic development [16]. Governments have increased investment in programs that promote an entrepreneurship mindset in education and society and make countries more entrepreneur-friendly [17–20]. Like many other countries, the Kingdom of Saudi Arabia (KSA) encourages undergraduate and graduate students to be involved in entrepreneurship and consider entrepreneurship as a career choice.

In the context of KSA, due to the growth of mass higher education, the employment of university graduates has become a major concern. According to the Saudi Central Department of Statistics and Information, the Saudi Arabian Monetary Agency, and the Saudi Ministry of Education, the unemployment rate is estimated to be 12.9%, while the number of undergraduate students from public and private higher educational institutions was 1,590,878 in 2021 [21]. On the other side, the Saudi youth, particularly undergraduates and fresh graduates, are less enthusiastic about working in the private sector and prefer to join the public sector. Most undergraduates and fresh graduates are looking for more stable jobs rather than starting their businesses in the face of rising unemployment and an increasing number of graduates each year, which makes it difficult to find a stable job [22]. A recent report 2021/2022 by the Global Entrepreneurship Monitor (GEM) [23] indicated that entrepreneurship activity rates are still at a low level in most countries worldwide compared to before the COVID-19 pandemic. However, there are expectations that KSA will experience an increase in entrepreneurship activities due to the growth over the past two years. According to the World Bank Doing Business Report 2020 [24], KSA has one of the world's largest economies, which makes it appropriate for starting a new business.

The Saudi Vision 2030 aims to make a transition in the national economy from full reliance on oil to a more diverse economy by localizing industries and encouraging entrepreneurship [22]. The Saudi Vision 2030 aims to reduce national youth unemployment from 12.9% to 7% [25]. Therefore, the government has been working at breakneck speed to establish a favorable entrepreneurship ecosystem. In this pursuit, the government has reformed rules and regulations, removed barriers, and increased access to financial institutions [25]. Hence, entrepreneurship is a crucial tool for supporting the Saudi Vision 2030. Consequently, the government has provided financial and regulatory support for universities across the kingdom to incorporate entrepreneurship more actively into their educational curricula. Colleges have a critical impact in promoting entrepreneurship education to support and inspire students to become self-employed once they have graduated [26]. This is especially true for disciplines such as agriculture and food science because they are an important part of the Saudi Vision 2030 to achieve food security for the kingdom.

Higher education students are expected to be successful entrepreneurs [27]. However, there needs to be a better understanding of the factors that lead students to become successful entrepreneurs. This subject necessitates extensive research and investigation, especially in the Saudi context, to support the national agenda. Hence, it is crucial to determine the factors that influence university students' intentions to become entrepreneurs. The important factor that is required to be embedded in a new start-up is entrepreneurship intention [28]. Entrepreneurship orientation is a behavioral and attitudinal variable that is expected to increase the predictability of a person's desire to become an entrepreneur [29]. Furthermore, a realistic understanding of students' entrepreneurship orientation will be critical in determining their eagerness to pursue an entrepreneurial career in the future [30]. Entrepreneurship orientation can be defined as a tendency to explore new business opportunities [31]. Entrepreneurship orientation has three dimensions: risk-taking, innovativeness, and pro-activeness [32].

Research has confirmed that the three dimensions of entrepreneurship orientation should be researched as different entities because they are each unique [33]. Do and Dadvari [34] revealed that the dimensions of entrepreneurship orientation were strongly related to university students' intentions to become entrepreneurs. Do and Dadvari [34] examined

the impact of various entrepreneurship orientation dimensions on entrepreneurship intention and suggested that, aside from innovativeness, risk-taking and pro-activeness were significant in directing students' intentions to pursue entrepreneurship activities as a future career choice. Thus, entrepreneurship orientation is a crucial indicator of development for potential entrepreneurs [35]. Additionally, Kreiser et al. [36] suggested that future studies should pay more attention to analyzing the interrelationships between the three sub-dimensions of entrepreneurship orientation. Understanding these interrelationships will help in better understanding the entrepreneurship intentions of graduates, hence, promoting entrepreneurship activity among graduates of higher education. This suggestion is supported by a meta-analysis on entrepreneurship orientation [33] that the connections between the three dimensions of entrepreneurship orientation require further investigation, which is considered in the current study. Al-Mamary et al. [37] claimed that a few studies had investigated the link between entrepreneurship orientation and entrepreneurship intention, particularly in the setting of KSA. Nevertheless, few research studies have examined entrepreneurship orientation and its effect on entrepreneurship intention in developing counties' contexts [38,39]. Additionally, dimensions of entrepreneurship orientation such as pro-activeness, innovativeness, and risk-taking have not been thoroughly examined separately in studies on entrepreneurship intention [40,41]. Moreover, the indirect effect of risk-taking on entrepreneurship intention through dimensions such as pro-activeness and innovativeness has not been examined to the best of the research teams' knowledge.

The purpose of the current study is to fill a gap in knowledge concerning the interrelationship between the dimensions of entrepreneurship orientation, i.e., risk-taking, pro-activeness and innovativeness, and entrepreneurship intention, especially in the Saudi context. This study has four key objectives. First, the study examines the direct impact of risk-taking on the entrepreneurship intention of Saudi university graduates, especially agriculture and food science graduates. Second, the research tests the direct impact of both pro-activeness and innovativeness on graduates' entrepreneurship intention. Third, the research examines the direct impact of risk-taking on the pro-activeness and innovativeness of agriculture and food science graduates. Fourth, it investigates the indirect impact of risk-taking on entrepreneurship intention through pro-activeness and innovativeness among agriculture and food science graduates. The current study establishes relevant implications for policymakers, higher education practitioners, and academics, particularly in KSA, on how to promote entrepreneurship intention among higher education students, especially agriculture and food science graduates, through risk-taking, pro-activeness, and innovativeness.

## 2. Theoretical Background and Hypothesis Development

### 2.1. Defining the Study Constructs

According to Al-Mamary et al. [37], entrepreneurship can be defined as the procedures of organizing, managing, and developing a venture for profit purposes. This process is motivated by a person's desire and ability to do this. Shane [42] described entrepreneurship as the process of identifying, evaluating, and exploring opportunities to create value through innovation to produce goods and services. Abu Bakar et al. [43] mentioned that the practice of entrepreneurship arises from the recognition of entrepreneurship capabilities and the exploitation of entrepreneurship opportunities, leading to the creation of new businesses. Entrepreneurship can be considered as a venture creation process by a person, an "entrepreneur", who is willing to take a risk, search for change, is never satisfied with the existing condition, and continually exploits opportunities to create value [17].

Mustikawati and Bachtiar [44] described intention as the inherent force capable of inspiring and motivating the individual to pay attention. To date, there is no standard definition or single way to measure entrepreneurship intention [45]. Nevertheless, it is the state in which people, both physically and mentally, manifest their desire to establish businesses or organizations [46]. Entrepreneurship intention is considered to be the strongest predictor of entrepreneurship behavior, which translates into entrepreneurship action, and

without it, no further entrepreneurship steps can be taken [47–49]. Thompson [50] defined entrepreneurship intention as a "self-acknowledged conviction by a person who intends to set up a new business venture and consciously plan to do so at some point in the future". Entrepreneurship intention reveals a person's desire, willingness, and readiness to pursue entrepreneurship as a career path and participate in entrepreneurship activities [49,51]. Academics have paid particular attention to the factors that can trigger a person's intention to start an entrepreneurial venture, and numerous factors have been investigated [17,51].

Based on entrepreneurship literature (e.g., [52,53]), entrepreneurship orientation has gained critical conceptual and empirical attention. Miller [32] introduced the fundamentals of entrepreneurship orientation. Miller [32] defined entrepreneurship orientation as "engages in product market innovation, undertakes somewhat risky ventures, and is first to come up with proactive innovations, beating competitors to the punch" (pp. 771). Based on Miller's [32] view, the construct of entrepreneurship orientation includes innovativeness, risk-taking, and pro-activeness. These three dimensions have become the foundation of entrepreneurship orientation and a key to entrepreneurship intention. This research draws on Miller's [32] approach and adopts the three dimensions to examine entrepreneurship orientation and its impact on entrepreneurship intention. Do and Dadvari [34] indicated that the construct of entrepreneurship orientation is connected with the intention of students toward entrepreneurship. The same authors found that risk-taking and pro-activeness are significant in directing students' intention to be an entrepreneur in the future [34].

Risk-taking is the propensity to act bravely rather than cautiously [54]. Lechner and Gudmundsson [55] indicated that risk-taking is associated with risk-return and trade-off, as well as the likelihood of a loss or uncertainty tolerance [56]. Risk-taking is described as the capability to take calculated yet bold actions, like venturing into a new business, experimenting with new sources of finances, and/or making important resource commitments to new ventures in the wake of uncertain environmental conditions [57]. Hence, Usman and Hashim [58] confirmed that individual entrepreneurs are influenced by risk-taking behaviors to enter into businesses that many would avoid.

Innovativeness was defined by Miller [32] as the propensity to engage in creativity and experimentation through the introduction of new goods or services. Furthermore, innovation is the ability to recognize and participate in business activities in creative and unusual ways [59]. Innovativeness is concerned with fostering and encouraging new ideas, experimentation, and creativity that will lead to the development of new services, products, or processes. Innovativeness is crucial for a business start-up and practice by the entrepreneur.

Pro-activeness can be defined as "expropriating as well as leveraging economic opportunities while also predicting and achieving needs of the market before they could be wasted or carried out by potential competitors" [37]. Pro-activeness is concerned with being the "first mover" and other measures targeted to securing and protecting market share, as well as a forward-looking viewpoint expressed in actions taken in expectation of demand in the future [32,60,61]. In another meaning, pro-activeness was described as predicting future market preferences and needs. It is a forward-thinking, opportunity-seeking approach characterized by the launch of innovative products and services [62].

### 2.2. Risk-Taking and Innovativeness

Innovation management encompasses activities intended to generate new ideas and translate these ideas into action to create new products or services for the market [63,64]. It is, by definition, uncertain in terms of its outcomes; nevertheless, a positive attitude toward risk-taking encourages the achievement of reasonable innovation [65]. Several studies have investigated the correlation between risk-taking and innovation, whether in entrepreneurship and leadership studies or creativity studies [66–69]. Based on a managerial standpoint, risk management requires substantial resources to be invested in activities that could be profitable but are risky due to uncertainty about the results that can be achieved [70]. Earlier research findings on creativity focus on risk-taking at the employee and team levels, em-

phasizing the significance of encouraging risk-taking to stimulate entrepreneurs' creativity and experimentation [71]. Innovation plays a significant role in reducing risk-taking by entrepreneurs, where innovation comes into play, transforming adventure into success [67]. Risk-taking propensity can debatably be considered a key to innovation as it promotes the development of new and uncertain ideas, motivating the allocation of human, time, and financial resources for implementation [67,70]. Miller [32] argued that risk-taking promotes entrepreneurship actions, e.g., becoming innovative. At the firm level, Joshi et al. [65] argued that organizations with stronger propensities for taking risks will demonstrate higher levels of innovation, whilst those with excessive propensities for taking risks will experience lower levels of innovation. To conclude, innovation might assist graduate students in supporting the creation of new ideas, testing out novel approaches to solving problems, using updated technological methods, and improving currently available goods or services in the field. Thus, they are eager to take risks to achieve a high return and generally tend to act "boldly" in risky situations. These discussions can be formulated into the following hypothesis:

**Hypothesis 1 (H1):** *Risk-taking has a positive effect on the innovativeness of higher education students.*

### 2.3. Risk-Taking and Pro-Activeness

According to Al-Mamary et al. [37], the successful entrepreneur should think outside the box and be a proactive person to launch new ideas or products to the market. Entrepreneurs should rethink being business owners by taking risks. This does not imply that taking risks entails starting a business without any strategies or a clear vision while simultaneously anticipating fantastic outcomes. Instead, entrepreneurship risk-taking requires proactivity and innovation to reduce the level of risk [72]. Pro-activeness is an opportunity seeking to anticipate the demand of markets in the future, which may or may not be related to the present [32,73]. Furthermore, according to Raghuvanshi et al. [74], entrepreneurs who practice pro-activeness involve enthusiasm and the ability of the business to meet future challenges and gain first-mover in the market. Joshi et al. [65] investigated the relationship between entrepreneurship orientation dimensions and suggested that higher levels of pro-activeness and proclivity for risk-taking are connected with higher levels of innovativeness; however, excessive levels of pro-activeness and proclivity for risk-taking are connected with lower levels of innovativeness. Based on these arguments, it could be proposed that:

**Hypothesis 2 (H2):** *Risk-taking has a positive effect on the pro-activeness of higher education students.*

### 2.4. Risk-Taking and Entrepreneurship Intention

According to Al-Mamary and Alshallaqi [25], risk is the subjective likelihood of systemic failure, potential loss, or any unfavorable natural occurrence of a terrible event when participating in an activity or working experience. The personality attribute of risk-taking impacts attitudes toward entrepreneurship intention [75]. Thus, many people are hesitant about becoming successful entrepreneurs for a variety of reasons, including the inherent risk of operating in the entrepreneurship sector of the economy [25]. According to Al-Mamary and Alshallaqi [25], risk-taking and entrepreneurship intention are strongly associated. A successful entrepreneur thinks outside the box and takes measured risks when launching a new service or product to the market [37]. This necessitates taking into account ambiguity's risks. An entrepreneur is driven to take chances in order to be successful and profit at the highest rate. Therefore, individuals with a higher risk-taking proclivity have a better chance of succeeding in entrepreneurship [76]. Al-Nashmi [77] indicated that entrepreneurs are more willing to take risks and work in ambiguous situations. Studies (such as [26,37,75]) confirmed that risk-taking could influence entrepreneurship intention positively. Hamdan [78] indicated that the potential to become an entrepreneur,

the willingness to accept risks, and the desire to start a business influence entrepreneurship intention. Individuals with a high tolerance for risk are typically more driven to engage in entrepreneurship than those with a low tendency to take a risk. Hence, they are less interested in engaging in entrepreneurship adventures [79]. Based on these arguments, it could be proposed that:

**Hypothesis 3 (H3):** *Risk-taking has a positive effect on the entrepreneurship intention of higher education students.*

### 2.5. Innovativeness and Entrepreneurship Intention

Innovation puts firms at the top of the competitive business market [80]. Innovation depends on a certain knowledge base, talent, and experience, in order to develop a novel idea, solution, or recommendation [81]. Aloulou [6] argued that achievement and innovation attitudes are important factors in entrepreneurship intention. Mandongwe and Jaravaza [82] argued that both risk-taking and innovativeness determine women's entrepreneurship intention. The findings of a study by Al-Mamary and Alshallaqi [25] confirmed that innovativeness and autonomy have a significant positive impact on an individual's intention to start a new venture. Hence, the government and universities should inspire students to be inventive and creative because most students want to work traditionally [37]. Innovativeness and a tendency to take risks have been identified by research as the most common factors that influence people's aspirations to become entrepreneurs [83,84]. Mueller [85] and Wagner [86] stated that there is a positive connection between innovativeness and entrepreneurship intention. According to Zampetakis et al. [87], innovation ability does not predict entrepreneurship intent if it is not accompanied by a proactive approach toward dealing with entrepreneurship. Therefore, the following hypothesis is proposed:

**Hypothesis 4 (H4):** *Innovativeness has a positive effect on the entrepreneurship intention of higher education students.*

### 2.6. Pro-Activeness and Entrepreneurship Intention

Proactivity is one of the primary success factors in creating and maintaining entrepreneurship ventures [88]. Research (e.g., [78,79]) has emphasized that the desire to be an entrepreneur depends on the ability to take risks and the skill to be innovative and proactive about business engagement. Earlier studies (e.g., [89,90]) found that proactiveness is an antecedent of innovativeness. Additionally, Begley and Boyd [83] stated that entrepreneurship intention is directly related to several entrepreneurship orientation dimensions, including pro-activeness, the ability to take risks, and innovation. According to Mustafa et al. [91], proactive individuals have a higher intention to be business owners compared to less proactive individuals. Therefore, it can be stated that proactive students are more likely to show entrepreneurship intention. Thus, the following hypothesis is suggested:

**Hypothesis 5 (H5):** *Pro-activeness has a positive effect on the entrepreneurship intention of higher education students.*

### 2.7. The Mediating Effect of Innovativeness and Pro-Activeness in the Link between Risk-Taking and Entrepreneurship Intention

Earlier studies (e.g., [26,37,75]) confirmed that risk-taking is a key factor that directly influences entrepreneurship intention. Mueller [85] and Wagner [86] suggested that there is a direct effect between innovativeness and entrepreneurship intention. Similarly, Mustafa et al. [91] indicated that proactive individuals have a higher intention to be an entrepreneur compared to less proactive individuals. According to Joshi et al. [65], most of the earlier studies treated the entrepreneurship orientation construct with its three dimensions (pro-

activeness, innovation, and risk-taking) as an aggregated construct, while Rauch et al. [33] indicated that there is a need to study entrepreneurship orientation dimensions as unique entities. Therefore, research (e.g., [36,92,93]) called to investigate the interrelationships between the three distinct entrepreneurship orientation dimensions. A study conducted by Joshi et al. [65] suggested that innovativeness has a curvilinear relationship with pro-activeness and risk-taking propensity. This study considers the first attempt to test the mediating effect of pro-activeness and innovations in the link between risk-taking and entrepreneurship intention. Hence, it could be hypothesized (as showed in Figure 1) that:

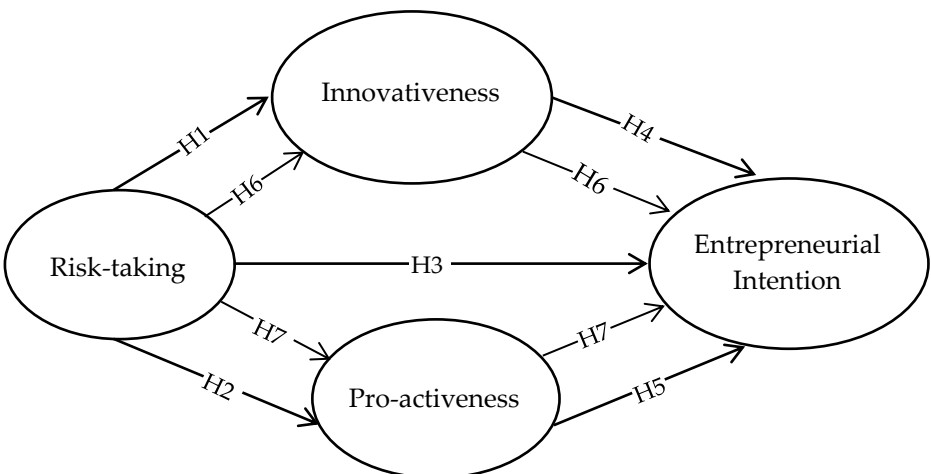

**Figure 1.** The theoretical model.

**Hypothesis 6 (H6):** *Pro-activeness has a mediating effect on the link between risk-taking and entrepreneurship intention of students in higher education.*

**Hypothesis 7 (H7):** *Innovativeness has a mediating effect on the link between risk-taking and entrepreneurship intention of students in higher education.*

### 3. Methodology

*3.1. Study Measures*

All of the multi-item scales used in this study were derived from a comprehensive evaluation and review of previously published empirical studies. This process produced four factors, each with its own set of items that were customized to the context of the study. The instruments were created using a seven-point Likert scale, where strongly disagree (1), disagree (2), somewhat disagree (3), either agree or disagree (4), somewhat agree (5), agree (6), and strongly agree (7). The entrepreneurship intention was operationalized by six items developed by Chen et al. [94] and Liñán and Chen [95]. Sample items include: "I am ready to do anything to be an entrepreneur". The scale showed a high Cronbach alpha (*a*) reliability with a value of 0.981. Similarly, entrepreneurship orientation was measured by three dimensions derived from Satar and Natasha [96]. The first dimension has three items measuring the graduate's ability to take a risk (risk-taking); an example item is: "I like to take bold actions by venturing into the unknown". The scale showed good reliability with a Cronbach alpha (*a*) value of 0.981. The second dimension has four variables and describes the graduate's innovativeness orientation. For example, the sample item: "I prefer to try my unique way when learning new things rather than doing it as everyone else does". The scale displayed good reliability with a Cronbach alpha (*a*) value of 0.960. Finally, the third dimension measures the graduate's pro-activeness ability. For example, the sample item: "I usually act in anticipation of future problems, needs, or changes". The scale has a high internal consistency with a Cronbach alpha (*a*) value of 0.971 [97].

During the pilot phase, thirteen business school professors and eleven graduates were given the questionnaire to guarantee its simplicity and consistency. The content of the questionnaire was not modified. The questionnaire asserts that the data collected is completely anonymous and strictly confidential. As the study questionnaire relies on a self-reporting collecting method, common method variance (CMV) may be a concern [42]. Harman's single-factor examination has been used to address CMV, with the extracted factors fixed to the value of 1.0 in an exploratory factor analysis (EFA) test that was performed through SPSS and the no rotation method. As only a single factor was obtained and explained 34% of the variance (less than 50%), CMV is not a concern [98].

### 3.2. Data Collection

All graduates enrolled in Saudi Arabian governmental institutions are the study population. Nearly 1 million college students were spread across 20 faculties in 2020. A random sample of 440 students enrolled in the fourth year at Saudi Arabia universities in the faculty of agriculture and food science was collected. The targeted graduates were taught a compulsory course at the university level during their undergraduate study entitled "Entrepreneurship Principles". The graduates were chosen to complete the questionnaire as they regularly think about their future careers and might possess an entrepreneurship orientation and intention. The questionnaire was dropped and collected from the targeted graduates in April and May 2022. The research team managed to disseminate 500 questionnaires, of which 450 answers were returned, while 10 questionnaires were eliminated due to incomplete data, resulting in a total of 440 valid questionnaires with a 90% response ratio. This sample size is satisfactory, according to Krejcie and Morgan [99]. Means for early and late responses were compared using the independent sample *t*-test method. The results indicated that non-response bias was not a concern in our study as no significant differences $p > 0.05$) were found between early and late responses mean [100].

### 3.3. Data Analysis Techniques

In this study, three different approaches were utilized in the process of data analysis: descriptive analysis (means and mode values, standard deviation, and skewness and kurtoses), confirmatory factor analysis (CFA) to test the scale's discriminant and convergent validity, and structural equation modeling (CBS-SEM) to evaluate the structural model hypotheses. Because it enables simultaneous investigation and evaluation of the complex interrelationships between multidimensional constructs, the CBS-SEM was chosen as the primary method for hypotheses testing. In addition, CBS-SEM can be used to investigate the relationships between variables while simultaneously taking into account the effects of measurement error. SPSS 25 and AMOS 24 (IBM, Armonk, NY, USA) were used for data analysis.

## 4. Findings

### 4.1. Descriptive Statistics

The vast majority (65%) of the respondents were male, and 85% were between the age of 18 to 23 years. Additionally, 35% of the graduates were from the King Faisal University (KFU) (located in the Eastern Province of KSA), 30% from Imam Mohammad ibn Saud Islamic University (IMSIU) (in the Riyadh Province of KSA), and 35% from Umm Al-Qura University (in the Makkah Province of KSA).

Table 1 displays the descriptive data of the responses to the study. The respondents' mean (M) values ranged from 4.10 to 5.36, and the standard deviation (S.D.) values ranged from 1.23 to 1.46, indicating that the results were more dispersed and less condensed around the mean value [101]. Table 1 provides the mode value of the most common answers of the study respondents, where the mode value of entrepreneurship intention was 0.5, which implies that the respondents somewhat agree with the variables that reflect entrepreneurship intention, while the mode value for risk-taking variables (5, 5, 6) imply that most respondents somewhat agree or agree to take the risk. Furthermore, the mode

value of three variable that describes pro-activeness (6, 6, and 6) imply that the study respondents agree with the statements that describe innovativeness. Finally, the mode value of the four variables of innovativeness (5, 5, 5, 4) indicates that respondents somewhat agreed or either agreed or disagreed with the sentences that reflect innovativeness. Table 2 also contains the skewness and kurtosis results of the data distribution, with no values exceeding −2 or +2, indicating that the data has a normal distribution curve [97].

**Table 1.** Descriptive statistics (mean, mode, standard deviation, skewness, and kurtosis).

| Abbreviation | Items | *M* | Mode | S. D | Skewness | Kurtosis |
|---|---|---|---|---|---|---|
| **Entrepreneurship Intention (Chen et al. [94]; Liñán and Chen [95]) (*a* = 0.981)** | | | | | | |
| E._Inten._1 | "I am ready to do anything to be an entrepreneur ". | 4.92 | 5 | 1.458 | −0.132 | −0.935 |
| E._Inten._2 | "My professional goal is to become an entrepreneur ". | 4.86 | 5 | 1.469 | −0.184 | −0.751 |
| E._Inten._3 | "I will make every effort to start and run my own firm". | 4.84 | 5 | 1.467 | −0.145 | −0.750 |
| E._Inten._4 | "I am determined to create a firm in the future". | 4.82 | 5 | 1.403 | −0.080 | −0.833 |
| E._Inten._5 | "I have very seriously thought of starting a firm". | 4.80 | 5 | 1.445 | −0.039 | −0.959 |
| E._Inten._6 | "I have the firm intention to start a firm someday". | 4.83 | 5 | 1.414 | −0.069 | −0.850 |
| **Risk-taking (Satar and Natasha [96]) (*a* = 0.960)** | | | | | | |
| Risk_1 | "I like to take bold actions by venturing into the unknown". | 4.84 | 6 | 1.40 | −0.162 | −0.932 |
| Risk_2 | "I am willing to invest a lot of time and/or money in something that might yield a high return". | 4.81 | 6 | 1.37 | −0.163 | −0.880 |
| Risk_3 | "I tend to act "boldly" in situations where risk is involved". | 4.79 | 5 | 1.26 | −0.050 | −1.113 |
| **Innovativeness (Satar and Natasha [96]) (*a* = 0.960)** | | | | | | |
| Innov_1 | "I often like to try new and unusual activities that are not typical but not necessarily risky". | 4.12 | 5 | 1.32 | −0.099 | −0.422 |
| Innov_2 | "In general, I prefer a strong emphasis in projects on unique, one-of-a-kind approaches rather than revisiting tried and true approaches used before". | 4.10 | 5 | 1.29 | −0.162 | −0.428 |
| Innov_3 | "I prefer to try my own unique way when learning new things rather than doing it like everyone else does". | 4.13 | 5 | 1.288 | −0.125 | −0.452 |
| Innov_4 | "I favor experimentation and original approaches to problem-solving rather than using methods others generally use for solving their problems". | 4.15 | 4 | 1.23 | 0.058 | −0.413- |
| **Pro-activeness (Satar and Natasha [96]) (*a* = 0.971)** | | | | | | |
| Proact_1 | "I usually act in anticipation of future problems, needs or changes". | 5.35 | 6 | 1.31 | −0.925 | 0.555 |
| Proact_2 | "I tend to plan ahead on projects". | 5.36 | 6 | 1.25 | −0.904 | 0.711 |
| Proact_3 | "I prefer to "step-up" and get things going on projects rather than sit and wait for someone else to do it". | 5.35 | 6 | 1.24 | −0.905 | 0.782 |

**Table 2.** Psychometric properties of the study scale.

| Factors and Variables | Standardized Factor Loadings | CR | AVE | MSV | 1 | 2 | 3 | 4 |
|---|---|---|---|---|---|---|---|---|
| 1—Entrepreneurship Intention | | 0.980 | 0.893 | 0.270 | **0.945** | | | |
| E._Inten._1 | 0.968 | | | | | | | |
| E._Inten._2 | 0.928 | | | | | | | |
| E._Inten._3 | 0.942 | | | | | | | |
| E._Inten._4 | 0.937 | | | | | | | |
| E._Inten._5 | 0.919 | | | | | | | |
| E._Inten._6 | 0.975 | | | | | | | |

**Table 2.** *Cont.*

| Factors and Variables | Standardized Factor Loadings | CR | AVE | MSV | 1 | 2 | 3 | 4 |
|---|---|---|---|---|---|---|---|---|
| 2—Risk-Taking | | 0.969 | 0.911 | 0.270 | 0.520 | **0.955** | | |
| Risk_1 | 0.971 | | | | | | | |
| Risk_2 | 0.970 | | | | | | | |
| Risk_3 | 0.922 | | | | | | | |
| 3—Innovativeness | | 0.960 | 0.858 | 0.168 | 0.410 | 0.400 | **0.926** | |
| Innov_1 | 0.902 | | | | | | | |
| Innov_2 | 0.935 | | | | | | | |
| Innov_3 | 0.933 | | | | | | | |
| Innov_4 | 0.935 | | | | | | | |
| 4—Pro-activeness | | 0.978 | 0.936 | 0.212 | 0.390 | 0.370 | 0.460 | **0.968** |
| Proact_1 | 0.958 | | | | | | | |
| Proact_2 | 0.971 | | | | | | | |
| Proact_3 | 0.974 | | | | | | | |

Model GoF: ($\chi^2$ (98, *n* = 440) = 390.762, *p* < 0.001, normed $\chi^2$ = 3.987, RMSEA = 0.072, SRMR = 0.0301, CFI = 0.950, TLI = 0.938, NFI = 0.941, PCFI = 0.776 and PNFI = 0.769). Note: CR: composite reliability; AVE: average variance extracted; MSV: maximum shared value; Bold diagonal values: the square root of AVE for each dimension; below diagonal values: inter-correlation between dimensions.

### 4.2. Results of Confirmatory Factor Analysis (CFA)

In order to evaluate the validity and reliability of the scale that was used, each of the independent and dependent factors, in addition to their respective reflective variables, were taken into consideration and subjected to Amos' first-order CFA and estimation method of maximum likelihood (MLE). As suggested by Hair et al. [101], Bryman and Cramer [102], Fornell and Larcker [103], Anderson and Gerbing [104], and Kline [105], various goodness of fit (GoF) measures were employed to test the model's fit to the gath-ered empirical data, including the chi-square scores divided by the degree of freedom (df) normed chi-square, Comparative Fit Index (CFI), Tucker Lewis index (TLI), and root means square error approximation (RMSEA). The GoF criteria of the tested model proved that the CFA has an acceptable and adequate fit to data (Table 2 and Figure 2). The scale reliability was assessed through two methods: Cronbach's alpha values (as previously ex-plained in the study measure section) and composite reliability (CR) scores. Results in Table 2 showed that CR scores for the four employed dimensions: entrepreneurship in-tention (0.980), risk-taking (0.969), innovativeness (0.960), and pro-activeness (0.978), ex-ceeded the cut-off value of 0.7 as recommended by Fornell and Larcker [103], implying that the data have a satisfactory internal consistency. The results of the CFA showed two pieces of evidence that guarantee the convergent validity of the employed measure: first, all of the loadings were above 0.90 with a highly significant *p*-value greater than 0.001, as shown in Table 2 and Figure 2. Table 2 and Figure 2 demonstrate that all of the factor loadings values are above the cut-off value of 0.50, falling between the ranges of 0.902 and 0.974 [100]. Second, the average variance extracted (AVE) scores for all employed four measures: entrepreneurship intention (0.893), risk-tak-ing (0.911), innovativeness (0.858), and pro-activeness (0.936), exceeded the value of 0.50, demonstrating sufficient and acceptable convergent validity (Kline, 2015 [105]).

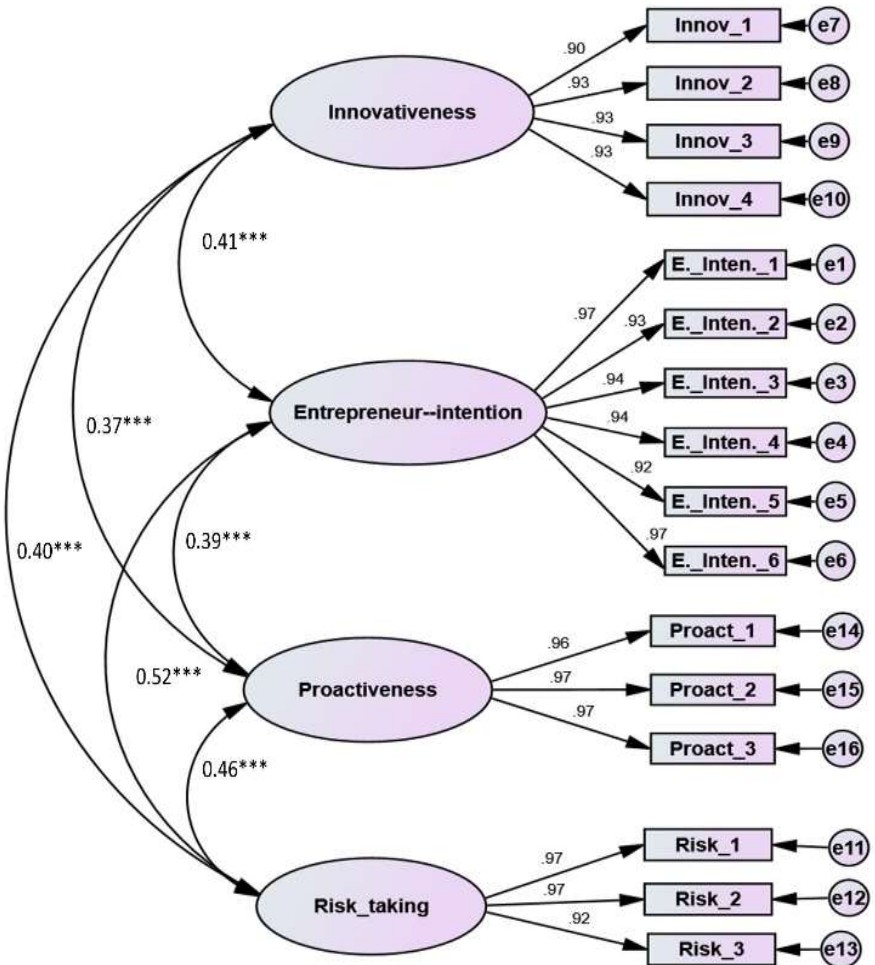

**Figure 2.** The CFA Model. Note: ***: significant level below 0.001.

Furthermore, the results of the CFA give two pieces of evidence as well that guarantee the discriminant validity of the employed measure, as suggested by Hair et al. [101], Bryman and Cramer [102], and Anderson and Gerbing [104]. First, the "maximum shared variance" (MSV) values should not be higher than AVE values, as displayed in Table 2. Second, for adequate discriminant validity, the squared root scores of the AVE values for the four employed measures (the bold diagonal data in Table 2) should be higher than inter-correlation (data below the bold diagonal scores), as represented in Table 2.

*4.3. Results of Structural Model*

In this research paper, we adopted a confirmatory strategy that consisted of two stages. In the first stage, a comprehensive literature review was conducted to develop the conceptual study model, and then observed data was collected in the second stage to decide if it matched the predesigned conceptual model [103]. The capability of the conceptual structural model to satisfy a model fit criteria (i.e., RMSEA, CFI, TLI, PCFI) was the basis for the evaluation that determined whether or not it should be approved. The GoF criteria gives evidence that model perfectly fit the observed data: $\chi^2$ (99, $n$ = 440) = 455.499, $p < 0.001$, normed $\chi^2$ = 4.601, RMSEA = 0.09, SRMR = 0.0375, CFI = 0.950, TLI = 0.940, NFI = 0.942, PCFI = 0.784 and PNFI = 0.777 (as displayed in Table 3). After determining whether or not the model adequately fit the data, the study's hypotheses were examined and evaluated. Figure 3 provides a visual representation of the hypotheses that have been proposed, where each individual path stands for a different hypothesis.

**Table 3.** The structural model's results.

| | Hypotheses | | | | Beta (β) | C-R (*t*-Value) | R² | Results of Hypotheses |
|---|---|---|---|---|---|---|---|---|
| H1 | Risk-Taking | → | | Innovativeness | 0.44 *** | 10.124 | | Supported |
| H2 | Risk-Taking | → | | Pro-activeness | 0.49 *** | 11.426 | | Supported |
| H3 | Risk-Taking | → | | Entrepreneurship Intention | 0.52 *** | 12.199 | | Supported |
| H4 | Innovativeness | → | | Entrepreneurship Intention | 0.39 *** | 8.870 | | Supported |
| H5 | Pro-activeness | → | | Entrepreneurship Intention | 0.42 *** | 9.901 | | Supported |
| H6 | Risk-Taking → Innovativeness → | | Entrepreneurship Intention | | Path 1: β = 0.44 *** *t*-value = 10.124 Path 2: β = 0.39 *** *t*-value = 8.870 | | | Supported |
| H7 | Risk-Taking → Pro-activeness → | | Entrepreneurship Intention | | Path 1: β = 0.49 *** *t*-value = 11.426 Path 2: β = 0.42 *** *t*-value = 9.901 | | | Supported |
| | Entrepreneurship Intention | | | | | | 0.60 | |

Model GoF: $\chi^2$ (99, *n* = 440) = 455.499, *p* < 0.001, normed $\chi^2$ = 4.601, RMSEA = 0.09, SRMR = 0.0375, CFI = 0.950, TLI = 0.940, NFI = 0.942, PCFI = 0.784 and PNFI = 0.777; ***: significant level less than 0.001.

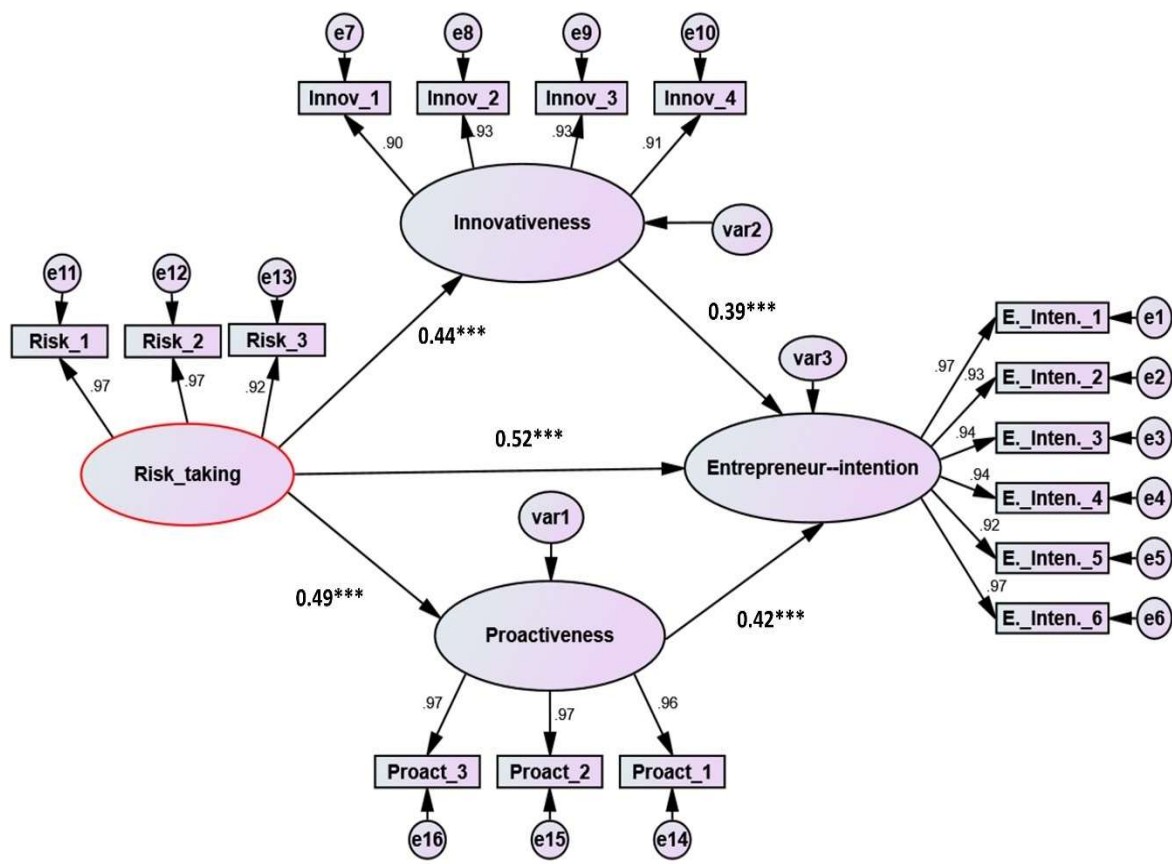

**Figure 3.** The Study Model. Note: ***: significant level below 0.001.

This study proposed seven hypotheses; five are direct, while two are indirect, as displayed in Table 3. The SEM output revealed that risk-taking positively and significantly impacts on entrepreneurship intention (β = 0.52. *t*-value = 12.199, *p* < 0.001), innovativeness (β = 0.44, *t*-value = 10.124, *p* < 0.001), and pro-activeness (β = 0.49, *t*-value = 11.426, *p* < 0.001) henceforth, the first three hypotheses, H1, H2, and H3 were accepted. Similarly, the results of the SEM showed that innovativeness has a positive and significant

impact on entrepreneurship intention (β = 0.39, *t*-value = 8.870, *p* < 0.001), supporting hypothesis H4. Likewise, pro-activeness was found to have significant and positive impacts on entrepreneurship intention (β = 0.42, *t*-value = 9.901, *p* < 0.001), hence, hypothesis H5 was supported.

Finally, the SEM results give evidence for the mediation effects of innovativeness and pro-activeness in the relationships between risk-taking and entrepreneurship intention. All the direct and indirect standardizer path coefficients in the tested model, as pictured in Figure 3, were found to be significant and positive; consequently, complimentary mediation is supported as recommended by Zhao et al. [106], and, thus, Hypotheses H6 and H7 can be accepted. Furthermore, the SEM output revealed additional indicators that confirm the mediation impacts on innovativeness and pro-activeness in the relationships between risk-taking and entrepreneurship intention, as the direct positive significant impacts of risk-taking on entrepreneurship intention was increased from (β = 0.52 *p* > 0.001) to a total effect of 0.63 with significant *p* > 0.001 [100]. Table 3 also demonstrates that the explanatory predictive power ($R^2$) of all paths ($R^2$ = 0.60) accounts for 60% of the variance in entrepreneurship intention.

## 5. Discussion and Implications

This research examined the effect of entrepreneurship orientation on entrepreneurship intention. More exactly, the research examined the direct effect of the three dimensions of entrepreneurship orientation: risk-taking, innovativeness, and pro-activeness on entrepreneurship intention. This research shows that risk-taking positively and significantly impacts innovativeness and pro-activeness. The research supports the previous literature review [65] that risk-taking encourages the achievement of innovation. Additionally, risk-taking propensity can debatably be considered a key to innovation as it promotes the development of new and uncertain ideas, motivating the allocation of human, time, and financial resources for implementation [67,70]. Furthermore, the current research also supports previous literature [37] that the successful entrepreneur should think outside the box and be a proactive person to launch new ideas or products to the market.

This research shows that risk-taking positively and significantly impacts entrepreneurship intention. This supports the work of Al-Mamary and Alshallaqi [25] that risk-taking and entrepreneurship intention are strongly associated. This is because entrepreneurs are more willing to take risks and work in ambiguous situations [77]. This is also supported by the work of Hamdan [78], who argued that the potential to become an entrepreneur, the willingness to accept risks, and the desire to start up a business influence entrepreneurship intention.

This research shows that innovativeness and pro-activeness have a positive and significant impact on entrepreneurship intention. This supports the work of Mueller [85] and Wagner [86], who found a positive connection between innovativeness and entrepreneurship intention. It also supports the results of Mustafa et al. [91] that proactive individuals have a higher intention to be business owners compared to less proactive individuals. However, it disagrees with Zampetakis et al. [87] that innovativeness does not predict entrepreneurship intent if it is not accompanied by a proactive approach toward dealing with entrepreneurship. The current research showed that innovativeness could predict entrepreneurship intention without the supplement effect of pro-activeness. This research confirms a mediating effect of both pro-activeness and innovativeness in the link between risk-taking and entrepreneurship intention. It was found that the effect of risk-taking on entrepreneurship intention increased with the availability of pro-activeness and innovativeness.

This research study has several theoretical and practical implications. Regarding the theoretical implications, this research study, for the first time to the best of researchers' knowledge, confirmed a complimentary mediation impact of pro-activeness and innovativeness in the link between risk-taking and entrepreneurship intention. These two dimensions of entrepreneurship orientation, i.e., pro-activeness and innovativeness, have the ability to enhance the effect of risk-taking on entrepreneurship intention among higher

education students, especially agriculture and food science graduates, who are the subjects of the current research. Hence, more emphasis should be directed by scholars to understand how pro-activeness and innovativeness can be ensured among students to enhance their entrepreneurship intention. Unlike the results of previous research studies [37,87], which confirmed no direct influence of innovativeness on entrepreneurship intention, the current research confirmed a direct and mediating effect of innovativeness on entrepreneurship intention. Additionally, unlike the work of Koe [4] that risk-taking is not an influential factor on entrepreneurship intention, the current research confirmed a direct and indirect effect of risk-taking on entrepreneurship intention. The three dimensions of entrepreneurship orientation have a positive and direct influence on the entrepreneurship intention of agriculture and food science graduates in KSA.

The current research also has some implications for higher education policymakers in relation to the design of entrepreneurship curriculum, which was recently added at higher education institutions in KSA universities. The curriculum should focus on the three dimensions of entrepreneurship orientation (risk-taking, innovativeness, and pro-activeness) to ensure the development of entrepreneurship intention among higher education graduates in KSA. Higher education policymakers need to pay special attention to developing the pro-activeness and innovativeness of graduates through both curriculum and training programs. This is because these factors were found to have a direct and mediating effect on the entrepreneurship intention of graduates. To promote innovativeness among graduates, the curriculum and training programs should encourage graduates to try new and unusual activities and try their own unique way when learning new things. The curriculum and training programs should also encourage students to favor experimentation and original approaches to problem-solving. Innovation should also be a core value of these graduates. Furthermore, to promote pro-activeness, the curriculum and training programs should encourage graduates to act in anticipation of future problems, needs, or changes and plan on projects. The development programs provided at the university should develop graduates' ability to be proactive. To support the Saudi Vision 2030 and its development programs, such as National Transformation Program and Human Capability Development Program, more attention is required regarding the development of entrepreneurship skills among agriculture and food science graduates, who are going to be entrepreneurs in this sector, which is crucial for food security in the kingdom. This can be achieved through quality entrepreneurship education and development programs while considering the factors that affect the entrepreneurship intention of these graduates.

## 6. Conclusions

The current research was concerned with examining the direct and indirect interrelationships of three dimensions of entrepreneurship orientation (risk-taking, pro-activeness, innovativeness) with the entrepreneurship intention of higher education students in KSA. KSA Vision 2030 aims to diversify the national economy by localizing industries and encouraging entrepreneurship. Vision 2030 aims to reduce youth unemployment from 12.9% to 7%. Consequently, the KSA government is moving quickly to create a favorable entrepreneurship ecosystem. Data were collected using a self-administrated questionnaire from 440 graduates in 3 main KSA universities (King Faisal University, Imam Mohammad ibn Saud Islamic University, and Umm Al-Qura University). CFA and SEM were employed to test the study measurement (for convergent and discriminant validity structure model) for hypotheses testing. The results showed that risk-taking as the exogenous variable positively and significantly impacts all the endogenous variables (innovativeness, pro-activeness, and entrepreneurship intention). Additionally, innovativeness and pro-activeness were found to partially mediate the relationship between risk-taking and entrepreneurship intention. These findings indicate the desire of agriculture and food science graduates to become entrepreneurs by being willing to accept risks, think outside the box, and be proactive in launching new ideas. Moreover, the findings show that pro-activeness and creativity can reinforce the impact of risk-taking on entrepreneurship intention.

The research was limited to students at one single public Saudi University "KFU". Hence, the results cannot be simply generalized to all universities in KSA or another context without further investigation of the current results. Additionally, the research did not examine the effect of demographics, e.g., gender of students, on the results of the current research, which could be an opportunity for further research.

**Author Contributions:** Conceptualization, I.A.E. and A.E.E.S.; methodology, I.A.E. and A.E.E.S.; software, I.A.E.; validation, I.A.E. and A.E.E.S.; formal analysis, I.A.E. and A.E.E.S.; investigation, I.A.E. and A.E.E.S.; resources, I.A.E.; data curation, I.A.E. and A.E.E.S.; writing—original draft preparation I.A.E. and A.E.E.S.; writing—review and editing, I.A.E. and A.E.E.S.; visualization, I.A.E. and A.E.E.S.; supervision, I.A.E. and A.E.E.S.; project administration, I.A.E. and A.E.E.S.; funding acquisition, I.A.E. and A.E.E.S. All authors have read and agreed to the published version of the manuscript.

**Funding:** This work was supported by The Saudi Investment Bank Scholarly Chair for Investment Awareness Studies, the Deanship of Scientific Research, Vice Presidency for Graduate Studies and Scientific Research, King Faisal University, Saudi Arabia (Grant No. CHAIR53).

**Institutional Review Board Statement:** The study was conducted according to the guidelines of the Declaration of Helsinki and approved by the deanship of the scientific research ethical committee, King Faisal University (project number: CHAIR53, date of approval: 1/February/2022).

**Informed Consent Statement:** Informed consent was obtained from all subjects involved in the study.

**Data Availability Statement:** Data is available upon request from researchers who meet the eligibility criteria. Kindly contact the first author privately through e-mail.

**Conflicts of Interest:** The authors declare no conflict of interest.

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
