# Peer review of "I Think I Can, I Think I Can: Effects of Entrepreneurship Orientation on Entrepreneurship Intention of Saudi Agriculture and Food Sciences Graduates"

_agriculture, doi:10.3390/agriculture12091454_

Round 1

Reviewer 1 Report

I appreciate the authors in evaluating the determinants of entrepreneurial intentions, especially in a sample that has not been thoroughly investigated. I would like to provide some suggestions to improve the current manuscript.

First of all, please use either “entrepreneurship intention” or “entrepreneurial intention” consistently throughout the whole manuscript.

Introduction:

This section is generally clear. However, some details can be explained further. For example, considering the sentence “Additionally, Kreiser et al., [36] suggested that future studies should pay more attention to analyzing the interrelationships between the three sub-dimensions of entrepreneurship orientation”, the authors can elaborate on the reasons for analysing entrepreneurship orientation. Considering the sentence “In this context, research related to entrepreneurship orientation is crucial”, the authors should also explain why analysing this is crucial.

Theoretical Background and Hypothesis Development:

Hypothesis development is generally clear and easy to understand. However, is there a theory which can serve as the theoretical foundation of the proposed model? Which theory/theories did the authors take account of when proposing this model? 

This research emphasises on “Agriculture and Food Sciences Graduates’ Perspective”. However, more details with related literature should be provided in supporting the reasons to analyse the agriculture and food science perspective. Are there any unique characteristics for this perspective which deserve further investigations? How would analysing the agricultural perspective enrich existing literature?

Data analysis:

Although the title indicates that the authors emphasise on “agriculture and food science perspective”, the sample comprised of graduates of other faculties and only 15% are from the faculty of agriculture and food science. The authors should clarify and explain the reasons to include graduates from other faculties, which may not be consistent with the title of this research. Notably, because this journal addresses agricultural issues or perspectives, the authors should elaborate on how this study connects with the scope of this journal. 

I recommend the authors to use “Standardized Loadings” or “Standardized factor Loadings” instead of “Standard Loading” in Table 2. 

Discussion and Implications:

Again, how did “agriculture and food science perspective” contribute to the significance of these hypotheses? Can the authors provide some reasons in connecting the significance of these hypotheses and the agricultural perspective? Any unique characteristics of graduates from in faculties of agriculture and food science may contribute to the significance of results?

Regarding the practical implications, the authors should provide more specific suggestions. For example, results reported that the effects of risk-taking, pro-activeness and innovativeness on entrepreneurial intention were prominent; what should governments and educators do in accordingly to enhance pro-activeness and innovativeness of students?

Conclusion:

The authors should add a section of conclusion to emphasise further on the major contributions of this research. Limitations can be incorporated as the final paragraph of conclusion.

Author Response

Dear Reviewer,

Thank you for giving us the opportunity to submit a revised draft of our manuscript titled “"I Think I Can, I Think I Can: Effects of Entrepreneurship Orientation on Entrepreneurship Intention of Saudi Agriculture and Food Sciences Graduates".” to agriculture journal. We appreciate the time and effort that you have dedicated to providing your valuable feedback on our manuscript. We are grateful to the reviewers for his insightful comments on our paper. We have been able to incorporate changes to reflect the suggestions provided in our revised manuscript. We have colored the changes within the manuscript in red.

Attached is a point-by-point response to your comments and concerns.

Reviewer 2 Report

The research is an extension of dozens of other similar researches on entrepreneurship orientation/ intention in different industries, contexts and countries. It is well-written and clear. It might add some value to the specific context referred to in the paper (KSA). Some suggestions for improvement follow:

- The theoretical model (figure 1) must also depict the hypotheses describing the mediating effects + bring some novel ideas/ dimensions; why do you need to test again something which is already validated in multiple researches before and it is obvious? The model should be extended to integrate a dimension/ particularity of KSA economy/ entrepreneurial environment. The authors should explain more clearly than it is actually presented why do they need to test the model in the selected economy and how they position their research in the existing literature.

- A discussion of the context and areas of the KSA economy requiring more entrepreneurial actions, and how, would help to justify the need for better entrepreneurship education. Concrete recommendations regarding entrepreneurship education are expected, as result of the research.

- The theoretical value of the research should be clearly outlined. What are the novel and unique contributions to the related literature? The paper should be more suggestive in terms of scientific relevance and contribution that the authors want to bring to the field. Implications on managerial practice should be also outlined. 

Author Response

(The authors gave the same response as above.)

Reviewer 3 Report

I read this manuscript "I Think I Can, I Think I Can: Effects of Entrepreneurship Orientation on Entrepreneurial Intention: Agriculture and Food Sciences Graduates' Perspective" with interest and care. Entrepreneurship is important for sustainable development and to decrease the pressure of employment on global public sectors as well as to lower unemployment rates. I have some suggestions which might assist the authors to increase the scientific quality of this article.

1.                  Please include country name in the title of the article.

2.                  Authors should also add sample size and sample method in the abstract section.

3.                  Page 2 paragraph 1: Authors should define all abbreviations used in the first place in the manuscript, like SME or SEM.

4.                  Page 2 paragraph 2: I think authors should use "under and fresh graduates" instead of "under or fresh graduates" in this paragraph.

5.                  Page 2 Section 3.1: Please define the 7-point Likert scale from 1 to 7. What does each number on the Likert scale indicate, from 1 to 7?

6.                  Page 8 Section 3.2: Please clarify about the population under study: was it post-graduate, graduate, or undergraduate students?

7.                  Can you explain a bit further the selection process of students for the Entrepreneurship course? Was it compulsory or selective? How many universities offered the course, and what was the selection process for the university/s? Was this course taught at the same time of the year to all students?

8.                  I would strongly recommend authors not to use the mean value of single statements in Table 1 (statistically it means nothing; please consult with a statistician for further information). I would recommend using one of the following: mode, median, or interquartile range for all individual statements (Table 1). Please amend the Table 1 description accordingly.

9.                  The authors should explain the SEM model in the methodology under a new section. Please also explain which SEM model you used, e.g., PLS SEM or CBS SEM.

10.              Section 4 should be named "Results or Findings" instead of "Data Analysis".

11.              The authors should write the discussion part without repeating the introduction and results of the study.

12.              Authors should also work on shortening the reference list because presently contains too many references (105).

Author Response

(The authors gave the same response as above.)

Round 2

Reviewer 1 Report

The authors have responded to my questions and concerns.

Reviewer 2 Report

Thank you for the improvements made. In the theoretical model you should distinguish between direct correlations and mediating effects. E.g.: use dotted lines.

Good luck with publication of your paper!

Reviewer 3 Report

Thank you for addressing my all comments.